# Methods for Studying Endometrial Pathology and the Potential of Atomic Force Microscopy in the Research of Endometrium

**DOI:** 10.3390/cells10020219

**Published:** 2021-01-22

**Authors:** Agnieszka Kurek, Estera Kłosowicz, Kamila Sofińska, Robert Jach, Jakub Barbasz

**Affiliations:** 1Jerzy Haber Institute of Catalysis and Surface Chemistry, Polish Academy of Sciences, Niezapominajek 8, 30-239 Krakow, Poland; 2Department of Gynaecology and Obstetrics, Jagiellonian University, Golebia 24, 31-007 Krakow, Poland; esteraklosowicz@gmail.com (E.K.); jach@cm-uj.krakow.pl (R.J.); 3M. Smoluchowski Institute of Physics, Jagiellonian University, Lojasiewicza 11, 30-348 Krakow, Poland; kamila.sofinska@uj.edu.pl

**Keywords:** endometrium, tissue properties, atomic force microscopy, AFM, endometriosis, cell, uterus

## Abstract

The endometrium lines the uterine cavity, enables implantation of the embryo, and provides an environment for its development and growth. Numerous methods, including microscopic and immunoenzymatic techniques, have been used to study the properties of the cells and tissue of the endometrium to understand changes during, e.g., the menstrual cycle or implantation. Taking into account the existing state of knowledge on the endometrium and the research carried out using other tissues, it can be concluded that the mechanical properties of the tissue and its cells are crucial for their proper functioning. This review intends to emphasize the potential of atomic force microscopy (AFM) in the research of endometrium properties. AFM enables imaging of tissues or single cells, roughness analysis, and determination of the mechanical properties (Young’s modulus) of single cells or tissues, or their adhesion. AFM has been previously shown to be useful to derive force maps. Combining the information regarding cell mechanics with the alternations of cell morphology or gene/protein expression provides deeper insight into the uterine pathology. The determination of the elastic modulus of cells in pathological states, such as cancer, has been proved to be useful in diagnostics.

## 1. Introduction

The global population of women has experienced an upsurge in the number of individuals suffering from endometrial diseases, particularly endometrial cancer [1,2]. Infertility, cancer, and endometriosis are the most common endometrial diseases. A tumor localized in the endometrium, particularly affecting women in developing countries, is the most common gynecological cancer and, among the most common cancers in women, its incidence has increased from fifth place in 2016 [3] to fourth in 2018 [1]. The endometrium tissue lines the uterus cavity and is composed of the epithelium layer, glands, and stroma cells [4]. Epithelial and stroma cells form two layers separated by a basement membrane that prevents epithelial cells from remaining in direct contact with the vasculature. Therefore, molecules circulating in blood, i.e., cytokines or hormones, only contact stromal cells, resulting in stroma cell activation. The basement membrane acts as a relay between stroma and epithelial cells. Thus, these two layers of the endometrium remain in constant, indirect contact. Stroma cells are essential for regulating epithelial cells’ growth and apoptosis, and presumably; only these cells are involved in the formation of pathological conditions such as endometriosis [5].

The most important role of the endometrium is related to the implantation of the embryo and to providing a place for its development and growth. The implantation process involves the attachment of the embryo to the endometrial surface of the uterus and the invasion of the epithelium to form the placenta [6]. This process is a crucial stage of fertilization, and is accompanied by several morphological rearrangements in the surface of the uterus lining, such as endometrium differentiation into decidua. Morphological features of the endometrium in the proliferative stage are crucial for successful implantation [7]. These structural and functional transitions are initiated and controlled by hormones, mainly progesterone and estrogen [6]. The process of implantation is usually divided into three main steps: apposition, adhesion, and invasion [6], but the initial interaction between the embryo and the endometrium may be featured as an additional step [8]. In the first step, the blastocyst contacts pinopodes located on epithelial surfaces of a receptive endometrium. Then, the adhesion of trophoblast cells of the blastocyst occurs. The final step is related to the penetration of the embryo through the epithelium to the endometrial stroma [6,8]. The cell-to-cell adhesion between trophoblast cells of the embryo and stromal cells of the endometrium is critical for the implantation [9,10]. Pathological changes regarding structural and mechanical properties of endometrial cells may be one of the causes of sterility [6].

The diagnosis of endometrial cancer is usually based on endometrial biopsies, and less often on a cervical smear test, but such an analysis does not often give a definite and unequivocal result [11]. The most characteristic feature of the endometriosis is the presence of stromal and epithelial cells outside the uterine cavity. Although endometriosis is a common gynecological lesion, its causes are not yet fully understood. Furthermore, typical endometriosis treatments appear to be inefficient [5]. Endometriosis is also considered a cause of infertility [12]. These facts confirm that further research on endometriosis is of high importance [5].

Due to the increasing number of individuals suffering from endometrial disorders, this tissue has been extensively studied during the past two decades. The main goal of the research published in this field has been related to the characteristics of endometrial cells and to understanding their functions. Numerous scientific efforts have also focused on the diagnostics of endometrial disorders and the elucidation of the origin of uterus disease formation or the occurrence of infertility. Despite the validity of endometrium research, endometrial tissue is still not fully understood, particularly in terms of its mechanical properties. The relationship of the mechanical properties of endometrial cells with the pathological states of the endometrium also needs to be developed and understood.

In this review, we present several techniques commonly used for endometrium characterization, in addition to non-conventional methods that are less known in the context of endometrium research. These non-conventional techniques may provide valuable information and have previously been successfully applied in the research of other cells and tissues. Table 1 summarizes methods previously used to characterize endometrial cells and tissue, and to understand diseases of the endometrium. We also provide examples of the research in which the described techniques were used. Methods described in Table 2 have previously been applied in the study of other cells and tissues. The mentioned techniques offer significant potential in the analysis of the endometrium. Among these techniques, atomic force microscopy (AFM) deserves special attention due to its possibility of combining information about the morphology of the cell with its mechanical properties.

## 2. Cell Morphology and Topography—First Source of Information

The information about cells morphology can be obtained using conventional microscopic techniques, such as the commonly used light microscopy or electron microscopy. Cells can be characterized and differentiated based on their ultrastructure features. Therefore, cell imaging is the main tool in cell biology.

The characteristics of endometrial cells requires the separation of cells into epithelial and stromal cells. The procedure developed by Kirk et al. enabled isolation and imaging of cells using electron microscopy techniques. Both transmission and scanning electron microscopy revealed structural differences between epithelial and stromal cells. The most characteristic features of the epithelial cell are the high regularity of the cell structure, cell junctions, and the presence of microvilli, which are absent in stromal cells. Techniques used for cell imaging proved the mesenchymal origin of stromal cells [14].

Fast and reliable disease diagnosis, and early detection of cancerous changes, provides better perspectives for recovery [36,37,38,39]. Correct classification of benign, atypical, and malignant cells is highly demanding but, nonetheless, essential for proper diagnosis. The developed and common procedures used for identification of pathological cells are based on direct endometrial samples. Liquid-based cytology (LBC) is not popular in this field, despite studies showing advantages of LBC samples. Gupta et al. reported a research concerning differentiation of exfoliated cells form cervical smears [13]. The aim was to compare morphological features of the nucleus, such as the nucleus area of LBC cells (Figure 1), and to show the usefulness of such analysis for sample classification. The analysis was carried out based on microphotographs obtained using an Olympus camera and opensource ImageJ software [40]. The described procedure allowed observation of significant morphological differences between benign and malignant cells regarding nucleus parameters such as area, mean diameter, mean perimeter, and standard deviation of the nuclear area. No differences were observed with respect to the mean integrated gray density. However, the presented procedure is not sufficient to clearly differentiate cells, especially atypical ones, so further studies are required in this field [13].

The main motivation for the investigation of human endometrial cells is their significant role in the implantation process. A common strategy concerning the research of endometrial cells is to compare endometrial samples from fertile and infertile women. TEM imaging allows observation of the morphological diversity between endometrial cells from fertile women and women with repeated implantation failure (RIF) associated with ultrastructures of the endometrial epithelium, such as cilia, pinopodes, and microvilli. Moreover, the modification of the endometrial cells observed during decidualization and endometrial secretion can be monitored. Bahar et al. indicated that pinopodes occurring in the endometrium of fertile women during the receptive period are essential for the success of the implantation process [7]. TEM also provided evidence that pinopodes are filled with the same light sensitive secretory material responsible for supplying nutrients to blastocytes, and possibly even allowing them to attach to the endometrium. The research of Bahar et al. confirmed the similarity in the size and number of microvilli in both fertile and RIF groups, but irregular structures with insufficient secretory production were detected in the RIF group (Figure 2). These studies have shown a strong correlation between infertility and the presence of cilia observed only in RIF samples. The authors suppose that the occurrence and movements of cilia limit the possibility of blastocytes attaching to the endometrium, making implantation impossible. During decidualization, the transformation of stromal cells into decidual cells in a group of fertile women was observed, whereas, in an RIF group, cell transformation was limited. Secretion is also limited due to a small area covered with glands. These two phenomena may also contribute to implantation failure [7].

Fluorescence phenomenon is commonly utilized in cell biology for the detection of proteins of interest or monitoring of gene expression. To capture a fluorescent response of molecules secreted or modified during the research, a confocal microscopy is usually used. This type of microscopy is often used in combination with other imaging methods, such as SEM or AFM, to complete the collected data with complementary information for confirmation that the studied processes had occurred [15,16]. The possibility of scanning of the sample in various planes makes confocal microscopy a helpful tool for 3D mapping that allows tracking of fluorescent molecules within the volume of the sample. Among proteins that can be detected using fluorescence staining, actin constitutes a common target for fluorescent microscopy. Actin is a building protein that maintains the cell structure and enables cell movement. The presence of actin is related to vinculin, the protein responsible for the proportion of actin F and G, and the formation of actin structures [41,42,43]. Wu et al. observed that differences in the expression of actin and vinculin result in changes of cell mechanical properties between endometrial cells isolated from healthy or endometriosis patients. This suggests that the endometriosis disease can be correlated with the mechanical properties of endometrial cells [5]. Pan-Castillo et al. used confocal microscopy supported by the fluorescence phenomenon to detect the increase in expression levels of vimentin, E-cadherin, and cytokeratin (Figure 3), which are markers of mesenchymal to epithelial transition, thus ensuring that the decidualization process occurred [15]. Another example is MUC1, a glycoprotein secreted under the influence of progesterone. The increase in MUC1 expression detected using confocal microscopy allowed confirmation that the cell is in a progesterone-dominant secretory phase of the menstrual cycle [16].

## 3. Cell Metabolism—Gene and Protein Expression

As mentioned above, protein and gene expression tracking allows correct classification of a cell state and detection of the processes running within the cell. Among the typical techniques used in biology or molecular biology, Western blotting is a common tool for detection and identification of proteins, and reverse-transcription polymerase chain reaction (RT-PCR) is a technique used for detection of nucleic acids (DNA, RNA), structural analysis, quantitative assessment, or determination of gene expression.

RT-PCR served as a prominent technique used in the research of *WT1* gene expression in the endometrium of infertile women suffering from polycystic ovary syndrome [17]. RT-PCR testing in combination with immunohistochemistry and Western blot tests confirmed a decrease in the expression of the *WT1* gene in the studied group, which was found to be connected with androgens. An elevated level of androgens is a symptom of polycystic ovary syndrome [17].

Techniques used for the detection of the level of gene expression supported with microscopic techniques can provide more consistent and reliable results. Francis et al. incorporated atomic force microscopy, confocal microscopy, RT-PRC, and Western blotting to observe the influence of progesterone on the modifications induced on the surface of the endometrial cells [16]. The changes in the nanoscale structure observed on the surface of endometrial cells need to be considered in conjunction with the increased expression of the *MUC1* gene and the MUC1 protein due to the relevance of the increased expression of MUC1 in the process of endometrium preparation to embryo implantation. Similarly, Wu et al. confirmed that changes in cell motility in patients with endometriosis are associated with the modification of Rho GTPase expression and activity [5]. This conclusion could only be reached due to the parallel Western blot analysis combined with the studies of cell mechanics. The expression level of Rho GTPase, which is a factor regulating the polymerization/depolymerization process of actin and controlling the occurrence of focal adhesion complexes, should be analyzed in conjunction with the studies of cell mechanical properties to push forward the state of knowledge concerned with endometriosis.

Another type of PCR (polymerase chain reaction) analysis is real-time PCR, which enables the observation of changes related to gene expression during subsequent cycles. Real-time PCR was used in the research of a decidualization process to demonstrate that the expression level of the Epidermal Growth Factor Receptor (EGFR) differed in decidual and untreated cells [15]. EGFR is a receptor tyrosine kinase that is responsible for regulation of various processes, including the functioning of the endometrium in the early stages of pregnancy [44,45]. To confirm that the decidualization process had occurred, protein blotting revealed changes in the expression of protein: vimentin was observed only in untreated cells, whereas E-cadherin and cytokeratin were present in decidual cells. The level of secretion of markers specific for the decidualization process, prolactin (PRL) and IGFBP-1 (insulin-like growth factor binding protein-1), have been determined by ELISA. Their secretion increased after 24 h from the induction of a decidual response. Therefore, the techniques mentioned above were used to confirm the ongoing processes, while providing valuable information about the processes taking place in the cell during decidualization [15].

Flow cytometry is a technique commonly used for quantitative assessment of cell populations, and for the analysis of morphological features and functional states. This technique is an important tool for cell phenotype determination and cell classification, e.g., after the isolation or the process of cell differentiation [21].

## 4. Molecular Spectroscopy and Nano-Spectroscopy

Raman spectroscopy is based on the phenomenon of inelastic scattering of monochromatic light. The difference in the energy of incident light and scattered radiation corresponds to vibrational frequencies characteristic of chemical bonds present in the studied molecules. Therefore, the Raman spectrum contains the information related to the chemical structure and composition of the studied material [12,46,47]. Raman spectroscopy is a rapid, non-invasive method allowing qualitative identification, in addition to quantitative analysis of proteins, nucleic acids, and lipids. Therefore, it is widely used in the research of biological samples (e.g., cells, tissues, blood) for the identification of chemical composition [48]. Parlatan et al. performed a non-destructive diagnosis of endometriosis using Raman spectroscopy [12]. Raman spectra of blood serum samples derived from healthy women and endometriosis patients (Figure 4a) were analyzed using principal component analysis (PCA) (Figure 4b,c), k-nearest neighbors (kNN), and support vector machines (SVM). Classification algorithms enabled an identification of endometriosis samples based on the Raman spectrum in the range of 1729–790 cm^−1^, without using a standard procedure of laparoscopy. The same bands, particularly 1156 and 1520 cm^−1^ attributed to C-C and C=C stretching vibrations of beta carotene, respectively, may be related to changes in the metabolism of retinoic acid and interpreted as potential markers of endometriosis. However, the significance of Raman band shifts and band intensity changes in the context of endometriosis diagnostics has not yet been entirely explained [12].

Another vibrational technique is infrared spectroscopy (IR), which, in combination with Raman spectroscopy, provides complete information on the chemical composition of the sample, due to different sensitivity with respect to vibrations of polar (IR) and polarizable groups (Raman) [22]. Due to the enhanced spatial resolution of Raman or infrared micro-spectroscopy, even less than 1 µm, it is possible to perform subcellular analysis. Notarstefano et al. studied endometriosis using Fourier transform infrared micro-spectroscopy (FTIRM) and Raman micro-spectroscopy (RMS) supported by PCA analysis. Granulosa cells (GCs) from both ovaries have been compared within three groups: GC samples from unilateral ovarian endometriosis (UOE) patients, from women with infertility of various origins (control group), and from the contralateral “healthy” ovary. Both IR and RMS spectra showed differences between GCs from the endometriosis patient and from the control group; however, no significant differences between two ovaries of endometriosis patients were observed. Spectral differences were observed in the whole spectrum. The most notable spectral changes were related to the intensity increase in stretching vibrations of =CH and C=O groups and the intensity decrease in CH_2_ groups, which correspond to the phenomenon of lipid peroxidation. Cellular oxidation that affects proteins and lipid may cause disorders in their functionality and is typical during endometriosis due to a reduced antioxidation ability of follicles. No differences in spectra of GCs from “healthy” and endometriosis ovarian cells suggest that oxidation disorders affect a larger area than the extent of the disease would indicate [22].

Tip-enhanced Raman spectroscopy (TERS) allows chemical mapping of the cell surface with a nanometric spatial resolution [35]. This technique combines chemical sensitivity of Raman spectroscopy with the resolution of atomic force microscopy or scanning tunneling microscopy (STM) [49,50,51]. This connection provides an enhanced spatial resolution of chemical recognition, efficiency, and sensitivity, which in the case of standard Raman spectroscopy may be insufficient, although it provides global information related to the studied samples. Böhme et al. applied TERS in the research of a biomembrane model composed of streptavidin-labelled phospholipid film [52]. TERS vibrational mapping allowed identification and analysis of the arrangement of molecules forming the studied biomembrane. Spectra obtained using TERS enabled determination of areas composed of protein, lipids, or both molecules simultaneously (Figure 5). Topography AFM images can be easily juxtaposed with TERS maps to translate the topography information into the chemical structure of a membrane (Figure 5b). Neugebauer et al. reported the first application of TERS in studies of complex biological systems related to the investigation of chemical composition and arrangement of components forming the surface *Staphylococcus epidermidis* bacteria [35]. Peptides and polysaccharides identified in this study were in agreement with a well-known composition of the *S. epidermidis* surface. Information collected using TERS may help to understand the processes occurring on the cell surface, such as cell adhesion, or the response for a drug treatment.

## 5. AFM in Studies of Physical, Chemical, and Mechanical Properties of Endometrial Cells

Atomic force microscopy (AFM) was originally intended to characterize semiconductor devices at a nanometric scale. It quickly became an important tool used in the analysis of biological matter due to the possibility of comprehensive analysis of living cells in addition to fixed material (cell, tissues, and cellular components such as proteins, DNA, or lipids). It is possible to perform AFM studies in the native biological environment, i.e., physiological buffers, or in ambient conditions (air) [53,54,55,56,57].

Several hormones, such as progesterone, may induce various modifications of cell morphological properties. In the case of progesterone, these differences are related to the glycocalyx thickness and a change of a surface charge. After progesterone treatment, an increase in cell height or a change in cell roughness is observed. Francis et al. showed differences in the surface roughness of endometrial cells over nuclear and cytoplasmatic areas [16]. Nuclear areas appeared to be characterized by higher values of surface roughness. A significant increase in the surface roughness was observed after progesterone treatment. AFM images enabled determination of the influence of external progesterone treatment on the roughness of endometrial cells at the micro- and nanoscale.

It was indicated by Bahar et al. that changes of morphodynamical features during endometrium decidualization can be crucial for implantation and maintenance of pregnancy [7]. Therefore, the understanding of transformations occurring in the decidualization process, being similar to a mesenchymal–epithelial transition (MET), is of high importance. Chemically induced decidualization with medroxyprogesterone acetate (MPA) and cAMP were studied using confocal microscopy to visualize cell structure, and AFM to imagine and quantify the roughness of the cell surface [15]. Morfometric analysis was performed with ImageJ software. The induction of decidualization has been established by following the level of two markers expressed during this process: decidual prolactin (dPRL) and IGFBP-1 (insulin-like growth factor binding protein-1). The morphological changes observed by Pan-Castillo et al. as a result of inducted decidualization were related to the increased roundness of cells, and to the increase in total cell area after 48 h. Using AFM high resolution peak force QNM (quantitative nanomechanical property mapping) mapping mode, it was possible to observe cell–cell and cell–glass surface interaction. AFM images indicated that neighboring cells were interacting through surface extensions stabilized by fibrillar cytoskeleton structures. Furthermore, stress fibers observed in control cells were thicker in comparison to fibers present in decidualized cells, whereas treated cells doubled their height. Moreover, a decrease in roughness coefficient was observed after decidualization. The results described above were supplemented with the indentation measurements using AFM, which did not show any significant differences in elastic modulus or adhesion as a result of the inducted decidualization [15].

Cell-to-matrix and cell-to-cell adhesion is a significant factor conditioning proper functioning of the body [24]. Adhesion disorders are a determinant of several diseases. In the case of endometrial tissue, the alteration of adhesion may contribute to implantation failure, and its changes may be observed in the cancer environment or during endometriosis. Integrins are glycoprotein receptors that are crucial to the adhesion process. The altered expression of integrins may be related to the presence of inflammatory cytokines that would explain the changes of adhesion found in endometriosis. Therefore, the examination and understanding of adhesion is one of the major goals of modern medical biology [58].

Micropipette aspiration is an important technique in the research of cell–substrate [24] and cell–cell adhesion [59]. Hogan et al. utilized micropipette aspiration to determine the cell–substrate adhesion force through a direct measurement of detachment force [24]. The procedure described on the example of bovine aortic endothelial cells allows determination of the influence of cell parameters (cell surface, adhesion area), and micropipette features (its size, loading rate) on the adhesion force. Optical tweezers (OT) may also be used to determine the adhesion force due to the possibility of cell displacement. Optical tweezers utilize the phenomenon of light refraction on small objects (e.g., cells) accompanied by a change in the momentum of photons. The momentum induces force acting on the object, enabling its manipulation [60,61,62]. Jing et al. showed that optical tweezers can be utilized to study cell adhesion and organization, using plasma-treated parylen-C film as a substrate for cell growth placed on photonic-crystal to enhance an optical trap above the substrate and to reduce photodamage [25]. Using this method, epithelial cells of the early embryo, which are pluripotent stem cells, were tested for intercellular adhesion while confirming that the novel system is a highly efficient and non-destructive tool for force measurements. Furthermore, the possibility of cell manipulation and colony formation has been demonstrated. Optical tweezers are also a useful tool in the studies of cell mechanical properties, such as stiffness. Yousafzai et al. used OT in the research of the influence of the presence of neighboring cells on the stiffness of breast cancer cells (selected results are summarized in Table 3) [26].

The adhesion force between the embryo and the endometrium is known to determine the success of the implantation process. The first report regarding this particular interaction was published in 1998 when Thie et al. began to study the adhesive forces occurring during implantation using atomic force microscopy [9]. This investigation was performed using model uterine epithelial cells, RL95-2 (RL) and HEC-1-A (HEC), of which the first exhibits adhesive properties to trophoblast-type cells (JAr), and the second does not show such a propensity. To determine the adhesion force between HEC or RL cells and JAr cells, a colloidal probe type AFM cantilever was covered with a monolayer of JAr choriocarcinoma cells (selected results are summarized in Table 4). It was found that the adhesion between RL or HEC and JAr cells does not occur immediately and increases with the extension of the contact time to about 20 min. Further extension of the contact time does not result in a significant increase in adhesion force. Such behavior may be related to the presence of multiple steps of cell-to-cell binding due to the signal transduction cascades. In contrast to HEC cells, the course of force curves collected for RL × JAr adhesion shows multiple rupture events during cantilever retraction. This suggests the appearance of specific interaction between RL and JAR cells. The breakage of this interaction leads to visible destruction of the RL monolayer. Based on approach curves, a transition from long-range soft repulsion to short-range hard repulsion forces was observed in the case of HEC cells during indentation, in contrast to RL cells. This is probably associated with differences of surface roughness: the rough HEC surface with microvilli and microridges, and the relatively smooth surface of RL cells, were demonstrated using TEM.

In addition to adhesion, other parameters can also be monitored as markers of pathological states. The mechanical properties of cells, cell migration, or cell tension may be followed in the course of diseases such as endometriosis. The comparison of endometrial stromal cells (ESC) from healthy woman and endometriosis patients may allow observed differences to be explained. It has been demonstrated that ESC from endometriosis patients shows a higher ability to migrate [5]. This dependence was determined based on images from time-lapse video-microscopy captured every 30 min for 24 h. Furthermore, based on immunofluorescence, the increase in actin and vinculin expression was detected in normal ESC in comparison to cells affected by endometriosis. These proteins are involved in cytoskeleton formation; therefore, their distribution represents a cell tension. Higher mobility and cell stiffening are probably important markers which may help in the diagnosis of endometriosis [5].

Cell motility may be also correlated with cell stiffness determined using AFM. These parameters are closely related to the properties of cellular components, the cytoskeleton and nucleus. In general, the decrease in cell stiffness is accompanied by the increase in cell motility. This may be explained by a higher capacity of a softer cell to deform, allowing it to move. However, not all research confirms this dependence, particularly in the case of cells exposed to pharmacological or genetical modifications. Such cells usually show higher motility with the increase in stiffness [63].

Harada et al. studied induced protein secretion (in this case interleukin-8 (IL-8)) in the response to mechanical stretch [18]. The tensile strength was induced using a vacuum-driven stretch system operated by a computer, and protein expression was monitored with ELISA and RT-PCR. It was observed that the secretion of interleukin-8 increases under the influence of a cyclic stretch, and the presence of progesterone significantly inhibits the stretch-inducted secretion of IL-8. These findings are extremely important due to the fact that the endometrium is a tissue subjected to continuous stretching. It is known that IL-8 is involved in the processes of uterus contractility that accompany physiological phenomena such as menstruation and implantation, but also abnormal phenomena, such as endometriosis and dysmenorrhea. For this reason, it is important to study the mechanism of the secretion of IL-8 and the methods of controlling it [18].

AFM allows the study of mechanotransduction and, more specifically, to track transmission of mechanical force through the cell, to observe the force-induced reaction of the whole cell or to study the subcellular response of cells [64]. Locating the AFM tip above the point of interest of the cell, e.g., the nucleus, allows the cell to be pressed with a well-defined nanonewton force, thus enabling observation of the mechanical response of the cytoskeleton. Such an approach enabled Hadjiantoniou et al. to determine that the reaction to local force is not homogeneous in the whole cell [27]. It was confirmed that a highly localized deformation occurs far from the point of force application.

Furthermore, AFM allows cell mapping in terms of its elastic properties. The distribution of Young’s modulus has been determined for both animal and human cells, for example, MMTV-PyMT mouse cells as a model of mammary gland cancer [28], primary human pulmonary artery endothelial cells [29,30,31], or the epithelial-like breast carcinoma cell line [32]. Elasticity maps selected from the above-mentioned works are shown in Figure 6. The distribution of the elasticity modulus reflects the heterogeneous structure of the cell. Stiffer or more elastic regions on the elasticity map correspond to the arrangement of actin filaments [33]. Grady et al. carried out research that aimed to determine the relation of cell elasticity to its ultrastructure [23]. Using AFM phase contrast imaging, it was demonstrated that, in the case of healthy cells, actin filaments have a significantly greater impact on the cell elasticity than microtubules. This was determined with an application of two different cytoskeletal destabilizers. However, the effect of the presence of microtubules on elasticity in cancer cells is significant [23].

Atomic force microscopy has served as an efficient tool to follow the elasticity changes of cancer cells. Several cancer cell types from human or murine cell lines have been studied in terms of their mechanical properties. AFM studies of cell elasticity proved that cancer cells are “softer” and display lower values of elastic modulus in comparison to healthy cells [65,66,67]. AFM also allows the elastic modulus to be determined and the mechanics of a single cell to be studied. Ketene et al. developed a model for progressive ovarian cancer and demonstrated significant differences in elasticity and viscosity of cells at early, intermediate, and late stages of ovarian cancer. The elasticity and viscosity of late-stage mouse ovarian surface epithelial cells (MOSE) were lower than those of cells at an early stage of cancer (values of elasticity and viscosity are summarized in Table 5) [34]. Analogical dependence of cell elasticity on the stage of cancer was also confirmed in the case of breast cancer by the sound touch elastography (STE) technique (Figure 7) [68]. The observations of Ketene et al. regarding elasticity and viscosity changes were compared with results obtained using confocal microscopy, which confirmed ultrastructural changes in the cytoskeleton manifested by the decreased concentration of actin fibers. Due to a large number of studies presenting the differences of stiffness displayed by healthy and cancer cells, it is suggested that the elastic modulus of cells can be a marker of cancer diseases and could significantly aid cancer diagnosis [63].

## 6. Conclusions

Techniques used to study the alternations of various properties of cells include methods typically used in biological research, such as RT-PCR and Western blot, or generally used and successfully adapted to cell research, such as SEM, TEM, and Raman spectroscopy. In this review, we emphasized the relevance of combining different techniques or using various methods simultaneously in cell research to achieve more reliable and complete information of cell processes or cell reactions to external treatments. In recent years, atomic force microscopy has gained significant interest in the field of cell research because it allows the study of cell topography (shape, size, surface roughness) and the mechanical properties of cells (elastic modulus, the response to indentation). The potential of this microscopy was also proved by combining it with Raman or IR spectroscopy to achieve TERS or infrared nano-spectroscopy, respectively. Such combination allows maps of chemical composition to be derived with the nanometric resolution of AFM.

Endometrial cells and tissue have been widely studied, resulting in a large quantity of information regarding normal endometrium functionality, in addition to changes in protein and gene expression, and morphological and mechanical modification related to several types of endometrial disorders, including endometriosis, cancer, and causes of infertility of different origins. However, the pathogenesis of uterine diseases and treatments have not been fully explored. The changes related to uterine pathology have only been correlated with the mechanical properties of endometrial cells to a limited extent. Further research in this area could provide additional insights regarding the cell behavior subjected to various endometrial pathologies. It could also help to propose new hypotheses concerning the mechanism of the formation of uterus diseases. Therefore, a significant amount of development remains in this research field.

For the foreseeable future, research on endometrium properties via atomic force microscopy will continue to focus on establishing the relationship between the mechanical properties of the endometrium and fertility. In particular, the correlation of Young’s modulus of the surface of endometrial cells with female fertility is of major interest. The correlation of this information with pathological changes, such as polyps or fibroids diagnosed in the uterus, may allow the future development of pharmacological methods to modify the mechanical properties of the endometrium to increase the probability of a successful pregnancy. Experimental research should be complemented by computer modeling. The recent progress of mathematical methods (high computing power with low time cost) will allow an understanding of the mechanisms that govern the ability of successful embryo implantation. Moreover, the significant development of medicine and the interdisciplinary research in this area will allow essential knowledge to be gained regarding the endometrium pathology and possible treatment strategies. At present, the mathematical description of the cell allows consideration of the cell microstructure and individual mechanical properties of the organelle. The use of the finite element method (FEM) allows the behavior of the cell under pressure to be determined, reflecting the measurements made via AFM. Further research on the mechanical properties of the endometrium will help to precisely diagnose infertility and endometrial pathology. It also appears that understanding the relationship between mechanical properties and fertility may lead to the development of new diagnostic procedures and methods, with reference to the historical foundations of medicine, such as physical examination of the abdomen (percussion and palpitations), but performed at the microscale.

## Figures and Tables

**Figure 1 cells-10-00219-f001:**
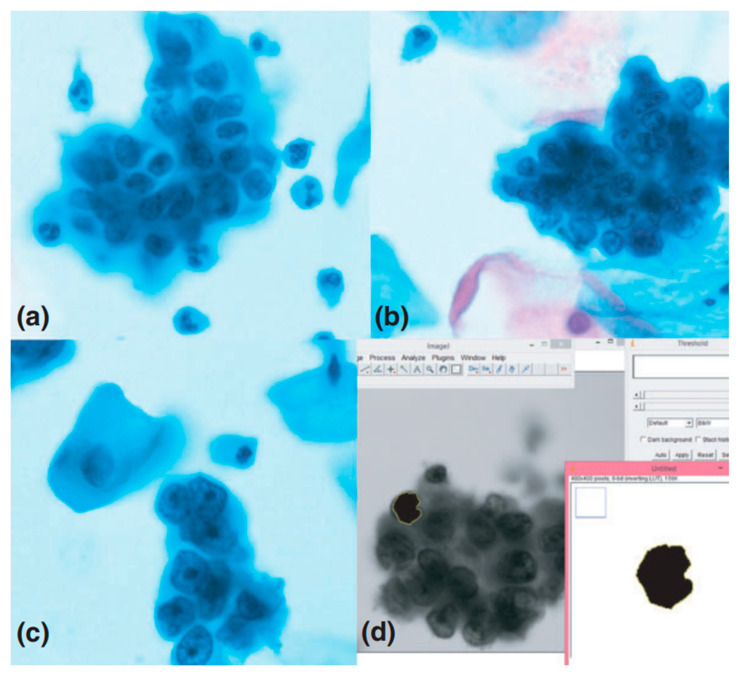
The comparison of nucleus morphological features to identify cancer cells. Microphotographs (1000× magnification) of liquid-based cytology (LBC) cervical samples showing (**a**) atypical, (**b**) benign, and (**c**) malignant cells, and (**d**) screenshot from ImageJ software used to determine the selected morphometric parameters of cells [13]. Reproduced with permission from Gupta, P.; Gupta, N.; Dey, P., Cytopathology; published by John Wiley & Sons, 2017.

**Figure 2 cells-10-00219-f002:**
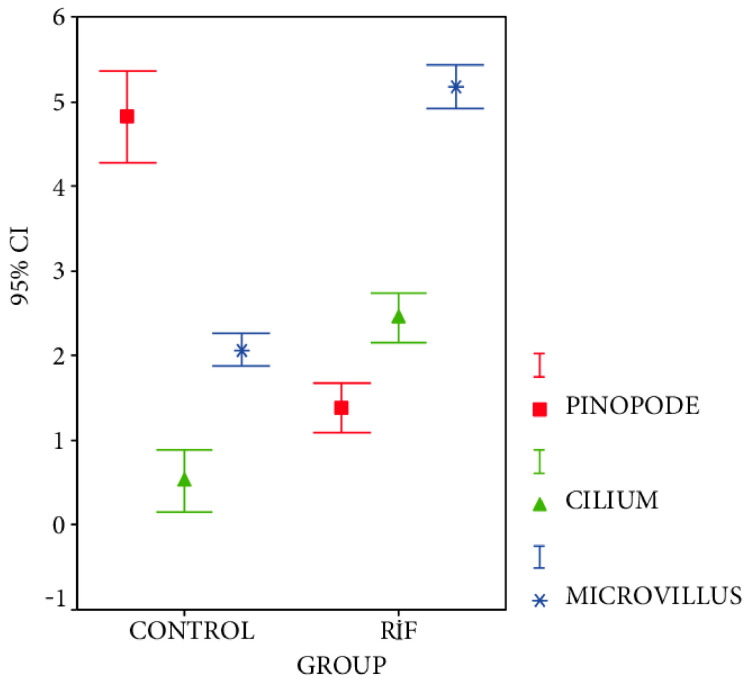
Comparison of cellular properties of endometrial tissues in the repeated implantation failure (RIF) group in reference to the control group. To identify differences between studied groups, the number of pinopodes, cilia, and microvilli has been analyzed [7].

**Figure 3 cells-10-00219-f003:**
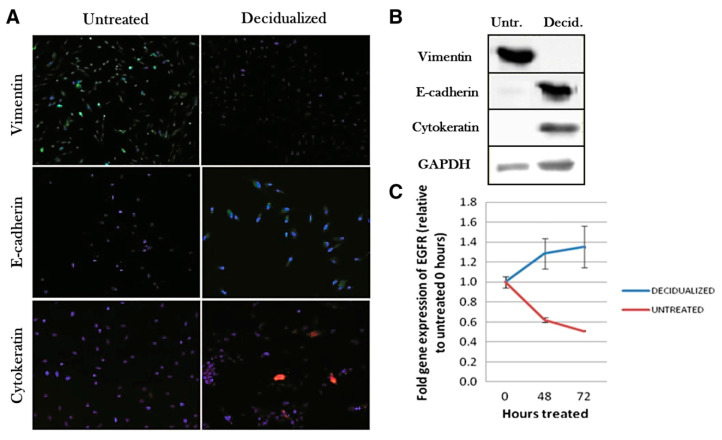
The expression of vimentin, E-cadherin, cytokeratin, and Epidermal Growth Factor Receptor (EGFR) in untreated and decidualized endometrial stromal cells (ESCs) studied to confirm mesenchymal to epithelial transition (MET). (**A**) confocal microscopy images demonstrating alterations in vimentin (green), E-cadherin (green) and cytokeratin (red) expression; (**B**) an example of Western blot technique utilization in the research of expression of vimentin, E-cadherin, cytokeratin, and GAPDH; (**C**) a graph of gene expression determined using the qPCR technique demonstrating the expression increase in EGFR in decidualized cells and the expression decrease in untreated cells [15].

**Figure 4 cells-10-00219-f004:**
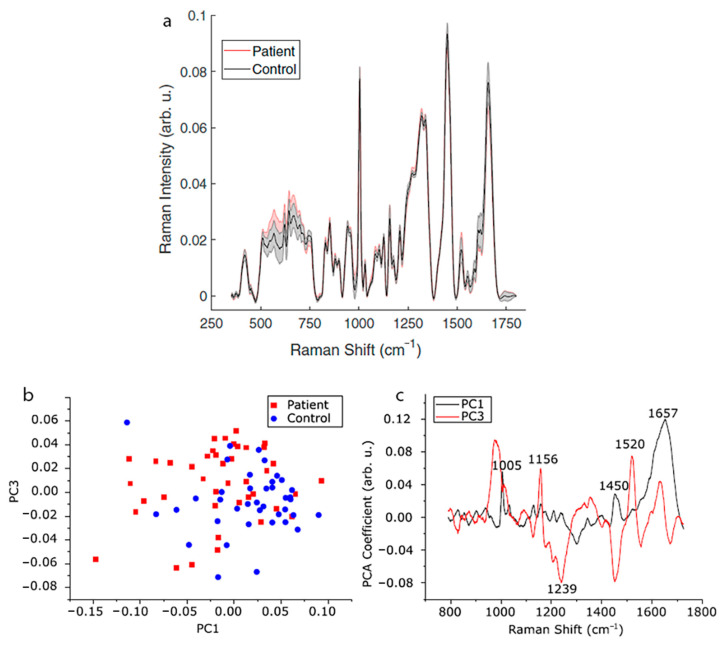
Raman spectroscopy combined with principal component analysis (PCA) analysis of spectra in the diagnosis of endometriosis. (**a**) Raman spectra acquired for endometriosis patients and the control group; (**b**) result of PCA analysis (PC1 vs. PC3) performed to demonstrate the differences between the studied groups; (**c**) shifts in the spectra that may correspond to chemical changes resulting from the disease [12].

**Figure 5 cells-10-00219-f005:**
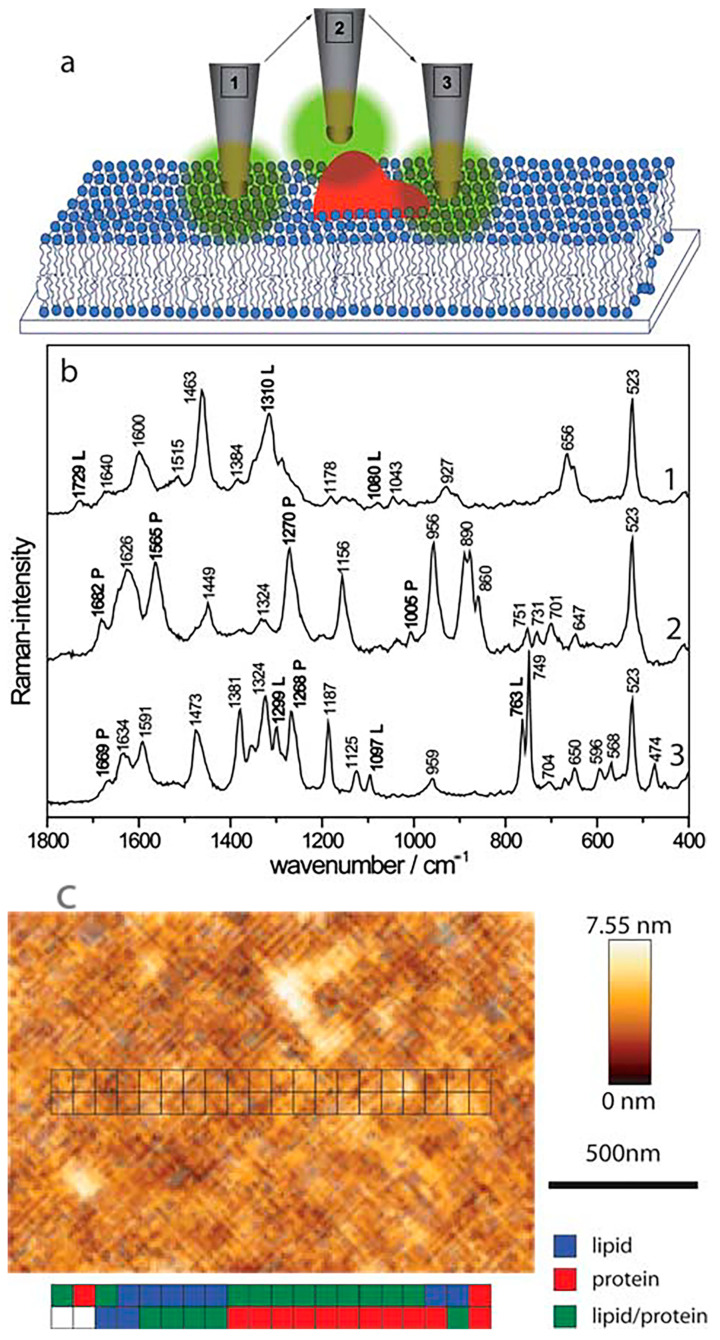
The application of the tip-enhanced Raman spectroscopy (TERS) technique in the analysis of the chemical composition of biomembranes. (**a**) a representation of possible atomic force microscopy (AFM) tip positions corresponding to the measurement of interactions with lipid domains (1), protein domains (2) or both domains simultaneously (3); (**b**) TERS spectra corresponding with tip positions presented above. Bands characteristic of protein and lipid domains are marked with “P” or “L”, respectively; (**c**) AFM topography image of streptavidin-labelled phospholipid film with marked positions of particular domains detected with TERS [52]. Reproduced with permission from Böhme, R.; Cialla, D.; Richter, M.; Rösch, P.; Popp, J.; Deckert, V., J. Biophotonics; published by John Wiley & Sons, 2010.

**Figure 6 cells-10-00219-f006:**
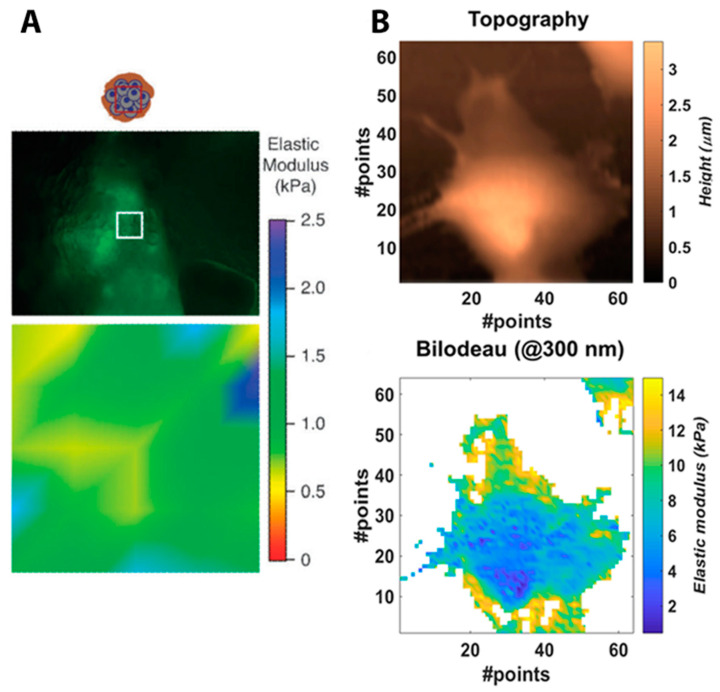
Mapping of mechanical properties of single cells using atomic force microscopy (AFM). Selected maps of elastic modulus present the elasticity distribution of a mammary gland tumor cell (indentation area—white square: 90 × 90 μm) (**A**) [28] Reproduced with permission from Lopez, J.I.; Kang, I.; You, W.K.; McDonald, D.M.; Weaver, V.M., Integrative biology; published by Oxford University Press, 2011. And a breast carcinoma cell (indentation area: 100 × 100 μm) (**B**) [32].

**Figure 7 cells-10-00219-f007:**
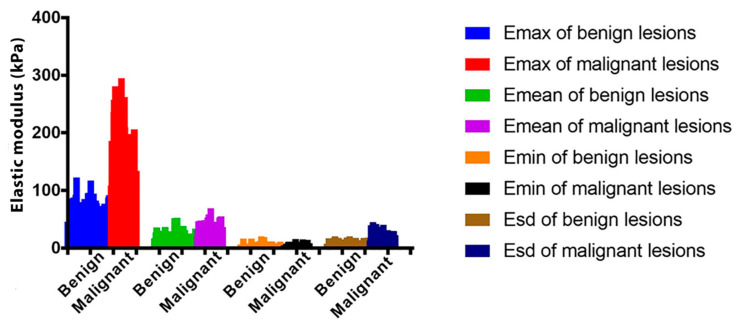
The differentiation of cancerous lesions based on elasticity measurements. Elastic modulus (E) of benign and malignant breast lesions determined by the sound touch elastography (STE) technique [68].

**Table 1 cells-10-00219-t001:** An overview of techniques used in the study of endometrial cells and tissues with examples of the research in which they were applied.

	Technique	Example of Application
Microscopic techniques	Optical microscopy	Classification of benign, atypical and malignant cells [13]
	Transmission and scanning electron microscopy	Analysis of structure differences between epithelial and stromal cells [14]
	Transmission electron microscopy	Observation of ultrastructural differences between endometrial cell from fertile woman and from woman with repeated implantation failure [7]
	Confocal microscopy	Molecules tracking as source of information on the course of cellular processes e.g., Observation of actin expression changes during endometriosis [5] Tracking of mesenchymal to epithelial transition markers [15] Analysis of phase of menstrual cycle [16]
	Atomic force microscopy	Height and roughness measurement, e.g., changes after progesterone treatment [16] or during decidualization [15]
	Time-lapse video-microscopy	Change of migration ability resulting from endometriosis [5]
Gene and protein expression analysis (both complete research and research support)	Reverse-transcription polymerase chain reaction and Western blot	Confirmation of the ongoing cellular processes based on the expression of gen or protein [5,16,17] Monitoring of stretch-induced protein secretion in the context of uterine contractility [18]
	Real-time polymerase chain reaction	Observation of gene expression changes during subsequent cycles e.g., in decidualization [15]
Enzymatic tests	Enzyme-linked immunosorbent assay	Analysis of surface markers e.g., in decidualization [15] Monitoring of stretch-induced protein secretion in the context of uterine contractility [18]
	Histochemistry	In situ activity analysis of ectonucleotidases as markers of endometriosis and endometrial cancer [19,20]
Analytical technique	Flow cytometry	Analysis of surface markers, assessment of cell quality, e.g., in cell phenotype determination [21]
Vibration technique	Raman spectroscopy	Chemical composition of sample, e.g., in endometriosis diagnosis based on Raman spectrum [12]
	Infrared micro-spectroscopy	Chemical composition of sample, e.g., changes caused by endometriosis [22]
Mechanical properties measurement	Atomic force microscopy	Determination of Young’s modulus and adhesion of decidualized cells [15] Determination of adhesion in implantation process [9]

**Table 2 cells-10-00219-t002:** An overview of techniques used in the studies of various cells and tissues characterized by high potential for the use in endometrial research together with examples of their applications.

	Technique	Example of Application
Microscopic techniques	Phase contrast microscopy	Observation of actin filament and microtubules impact on human umbilical vein endothelial cells, chondrocytes, fibroblasts, fibrosarcoma and hepatocellular carcinoma cells [23]
Mechanical properties measurement	Micropipette aspiration	Determination of bovine aortic endothelial cells adhesion [24]
	Optical tweezers	Intercellular adhesion of the early embryo epithelial cells determined by cell displacement [25] Assessment of the influence of neighboring cells on the stiffness of breast cancer cells [26]
	Atomic force microscopy	Detecting fibroblast cell inhomogeneities based on point cell pressing [27] Force mapping: Young’s modulus maps e.g., of mammary gland cancer [28], primary human pulmonary artery endothelial cells [29,30,31], epithelial-like breast carcinoma cells [32] and elasticity maps e.g., of human aortic endothelial cells [33] Elasticity measurements of human umbilical vein endothelial cells, chondrocytes, fibroblasts, fibrosarcoma and hepatocellular carcinoma cells [23] Differentiation of early, intermediate and late cancer stage base on elasticity and viscosity measurement [34]
Combination with atomic force microscopy	Tip-enhanced Raman spectroscopy	Chemical composition of sample with higher resolution, e.g., peptides and polysaccharides on bacteria surface [35]

**Table 3 cells-10-00219-t003:** Selected values of the Young’s modulus for luminal breast cancer, and normal and myoepithelial cells lines measured in three areas (isolated or contacted cells) [26].

	Elastic Modulus E (Pa) of Luminal Breast Cancer Cells MCF-7		Elastic Modulus E (Pa) of Normal and Myoepithelial Cells HBL-100	
	Isolation	Contact	Isolation	Contact
Region above nucleus (L1)	39 ± 8	20 ± 11	36 ± 11	30 ± 11
Cytoplasm in intermediate position (L2)	21 ± 10	18 ± 11	27 ± 10	25 ± 9
Region near the leading edge (L3)	16 ± 6	14 ± 7	19 ± 8	21 ± 8

**Table 4 cells-10-00219-t004:** Selected results of adhesion measurements between HEC-1-A (HEC) or RL95-2 (RL) cells and trophoblast-type (Jar) cells attached to a colloidal probe cantilever [9].

Measurements with JAr Covered Microbeads	HEC-1-A Cells	RL95-2 Cells
Distance range for soft repulsion observation	4.0 ± 0.3 μm	3.4 ± 0.4 μm
Indentation forces during the start of hard repulsion	1.0 ± 0.2 nN	0.7 ± 0.2 nN
Adhesive maximum after one minute contact	7.1 ± 2 nN	4.2 ± 2 nN
Adhesive maximum after one minute contact	16 ± 4 nN	-

**Table 5 cells-10-00219-t005:** Elasticity and viscosity properties of mouse cancer cells in early, intermediate, and late stages of cancer [34].

	Early Stage of Cancer	Intermediate Stage of Cancer	Late Stage of Cancer
Elastic modulus peak maximum (kPa)	0.652	0.477	0.382
Average elastic modulus (kPa)	1.097 ± 0.682	0.796 ± 0.441	0.549 ± 0.281
Viscosity rate peak maximum (Pa∙s)	69.69	60.40	25.53
Average Young modulus (kPa)	0.554 ± 0.349	0.472 ± 0.306	0.395 ± 0.136

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
