# Peer review of "Methods for Studying Endometrial Pathology and the Potential of Atomic Force Microscopy in the Research of Endometrium"

_cells, 2021, doi:10.3390/cells10020219_

Round 1
Reviewer 1 Report
The manuscript is not a review; it is very confused, jumping from different argument without a a logical thread, with many mistake starting from the most severe: "implementation" instead of implantation which is repeated several time through all the manuscript. This suggests few knowledge on the field of the review proposed by the Authors.
Author Response
Thank you for your time and criticism. We have replaced the word "implementation" with the word "implantation" in adequate lines. Please see the following lines: 44, 50, 52, 58, 123, 141, 305, 325, 355, and Table 1.
We have also improved our spelling and scientific wording. Additionally the language has been revised with MDPI English Editing Services.
Reviewer 2 Report
Overall, this work highlight the importance of current clinical issues and it is worth to published.
Author Response
We would like to thank for the positive review.
The English language and style have been additionally revised.
Reviewer 3 Report
In the manuscript entitled “A review of endometrial pathology studies and the potential of atomic force microscopy in the research of endometrium” by Kurek et al., the authors compile the state of art of endometrial research approaches.
The research of endometrium is of interest since it involves a number of pathologies, malignant, such as cancer, and non-malignant, but highly prevalent, such as endometriosis. Infertility also usually involves endometrial alterations. The authors provide a detailed information. There are some aspects that have to be improved before publishing.
- English should be extensively edited. Scientific writing should be edited as well. For example, avoid capitalizing words such as Confocal Microscopy, Interleukin, and many others. Please revise it throughout the text. Revise also the gene and protein names and their acronyms. Italics for genes.
- In order to help the reading, I suggest to split up the chapter 3 into two, one devoted to “cell metabolism-gene and protein expression” and the other to “spectroscopy”.
- In table 2, in “enzymatic tests” I suggest to include in situ histochemistry, since it is a technique used to characterize ectonucleotidases enzymes in endometrium, both pathological and non-pathological. There is a number of references in the literature.
- In conclusions, line 448, please replace “research article” by “review”.
- I suggest to include a paragraph exposing, based on the information provided in the text, specific proposals for future endometrial studies using AFM and how do the authors envisage the translation to the clinical practice.
Author Response
We would like to thank for the review.
- English should be extensively edited. Scientific writing should be edited as well. For example, avoid capitalizing words such as Confocal Microscopy, Interleukin, and many others. Please revise it throughout the text. Revise also the gene and protein names and their acronyms. Italics for genes.
Response: We have revised our spelling, scientific wording, and capitalizing words (a detailed list of changes is provided below). We also have applied italics for genes. Additionally the language has been revised with MDPI English Editing Services.
List of changes:
The change from the uppercase to the lowercase in the following words: “interleukin” (lines: 393, 396); “confocal microscopy” (lines: 148, 152, 160, 165, 190 and Table 1); “atomic force microscopy” (lines: 19, 26, 84, Table 1., Table 2., lines: 189-190, 265, 290, 355, 423, 426, 455); “scanning tunnelling microscopy” (lines: 265-266); “optical microscopy” (Table 1.); “transmission electron microscopy” (Table 1.); “time-lapse video-microscopy” (Table 1.); “reverse-transcription polymerase chain reaction” (Table 1.); “Western blot” (Table 1.), “real-time polymerase chain reaction” (Table 1.); Enzyme-linked immunosorbent assay (Table 1.); “Flow cytometry” (Table 1.); “Raman spectroscopy” (Table 1, line 244-245); “Raman micro-spectroscopy” (line 250), “cells” (line 251); “tip-enhanced Raman spectroscopy” (line 263); “infrared micro-spectroscopy” (Table 1., line 247); “Fourier transform infrared micro-spectroscopy” (lines 249); “phase contrast microscopy” (Table 2.); “micropipette aspiration” (Table 2.); ”optical tweezers” (Table 2.); “tip-enhanced Raman spectroscopy“(Table 2.); “human” (Table 2.); ”hepatocellular” (Table 2.); ”primary human pulmonary artery endothelial cells” (Table 2., lines: 412-413); “sound touch elastography” (line 436, 447-448); “mesenchymal to epithelial” (line 170); “polycystic ovary syndrome” (lines: 184, 187); “k-nearest neighbours” (line 230); “support vector machines” (line 230-231); “peak force” (line 315).
We have replaced the word "implementation" with the word "implantation" (lines: 44, 50, 52, 58, 123, 141, 305, 325, 355, and Table 1)
- In order to help the reading, I suggest to split up the chapter 3 into two, one devoted to “cell metabolism-gene and protein expression” and the other to “spectroscopy”.
Response: Thank you very much for this advice. Now we have two separate sections: “3. Cell Metabolism - Gene and Protein Expression“ starting in line 176, and section “4. Molecular spectroscopy and nano-spectroscopy“ starting in line 219.
- In Table 2, in “enzymatic tests” I suggest to include in situ histochemistry, since it is a technique used to characterize ectonucleotidases enzymes in endometrium, both pathological and non-pathological. There is a number of references in the literature.
Response: In Table 2, in “enzymatic tests” section we have included in situ histochemistry and we have added relevant references.
- In conclusions, line 448, please replace “research article” by “review”.
Response: “research article” was replaced with the word “review” (please see lines
452-453)
- I suggest to include a paragraph exposing, based on the information provided in the text, specific proposals for future endometrial studies using AFM and how do the authors envisage the translation to the clinical practice.
Response: We have included the additional paragraph in the section “6. Conclusions”. Please see line 472:
“For the foreseeable future, research on endometrium properties via atomic force microscopy will continue to focus on establishing the relationship between the mechanical properties of the endometrium and fertility. In particular, the correlation of Young's modulus of the surface of endometrial cells with female fertility is of major interest. The correlation of this information with pathological changes, such as polyps or fibroids diagnosed in the uterus, may allow the future development of pharmacological methods to modify the mechanical properties of the endometrium to increase the probability of a successful pregnancy. Experimental research should be complemented by computer modelling. The recent progress of mathematical methods (high computing power with low time cost) will allow understanding of the mechanisms that govern the ability of successful embryo implantation. Moreover, the significant development of medicine and the interdisciplinary research in this area will allow essential knowledge to be gained regarding the endometrium pathology and possible treatment strategies. At present, the mathematical description of the cell allows consideration of the cell microstructure and individual mechanical properties of the organelle. The use of the finite element method (FEM) allows the behavior of the cell under pressure to be determined, reflecting the measurements made via AFM. Further research on the mechanical properties of the endometrium will help to precisely diagnose infertility and endometrial pathology. It also appears that understanding the relationship between mechanical properties and fertility may lead to the development of new diagnostic procedures and methods, with reference to the historical foundations of medicine, such as physical examination of the abdomen (percussion and palpitations), but performed at the microscale.”
Round 2
Reviewer 1 Report
The review is still confused, without a focus well characterize. Different topics are still collected all together ( embryo implantation, endometrial cancer, endometriosis). This puzzle is still present; the Authors did not change the organization of the review: they only corrected single words, respect to the first submission. Therefore I still suggest "rejection"
The confusion is the main problem of this review, that in my opinion is not a review but a collection of different sections without any idea, first of all on the endometrial morphology and physiology.
Author Response
Dear Reviewer,
Thank you for your time and criticism. Since the title of our article was the main reason of the confusion, we decided to change it, because the review does not address the studies on endometrial pathology in depth. Our intention was to gather various methods used for studying endometrial pathology with the emphasize of the potential of the research on mechanical properties of endometrial cells in endometrial pathology. Therefore, the current title is: “Methods for Studying Endometrial Pathology and the Potential of Atomic Force Microscopy in the Research of Endometrium”.
Yours sincerely,
Agnieszka Kurek
Jakub Barbasz
